# Practical Low-Rank Communication Compression in Decentralized Deep Learning

**Thijs Vogels**
EPFL
thijs.vogels@epfl.ch

**Sai Praneeth Karimireddy**
EPFL
sai.karimireddy@epfl.ch

**Martin Jaggi**
EPFL
martin.jaggi@epfl.ch

## Abstract

Lossy gradient compression has become a practical tool to overcome the communication bottleneck in centrally coordinated distributed training of machine learning models. However, algorithms for decentralized training with compressed communication over arbitrary connected networks have been more complicated, requiring additional memory and hyperparameters. We introduce a simple algorithm that directly compresses the model differences between neighboring workers using low-rank linear compressors applied to model differences. Inspired by the PowerSGD algorithm for centralized deep learning (Vogels et al., 2019), this algorithm uses power iteration steps to maximize the information transferred per bit. We prove that our method requires no additional hyperparameters, converges faster than prior methods, and is asymptotically independent of both the network and the compression. Out of the box, these compressors perform on par with state-of-the-art tuned compression algorithms in a series of deep learning benchmarks.
This paper's code is available at `https://github.com/epfml/powergossip`.

## 1 Introduction

The major advances in machine learning in the last decade have been made possible by very large datasets collected by multifaceted organizations. We live in a society where almost every individual owns electronic devices that collect huge amounts of data, which—when used collaboratively—could lead to transformative insights (Nedic, 2020). Often this data is bound to the device it is captured on. This might be for practical reasons of communication efficiency, or for more fundamental reasons such as privacy constraints.

Decentralized machine learning enables collaborative processing of this new kind of data. In this paradigm, devices (nodes) have their own local data. The nodes joinly train a model by minimizing a loss function on their joint dataset. To do so, nodes communicate in a peer-to-peer fashion without any central coordination. A node can only communicate with few 'neighbor' nodes. This decentralized approach is not only useful in fundamentally decentralized systems, but the sparse communication patterns can sometimes even lead to efficiency gains in a datacenter (Assran et al., 2019).

In bringing decentralized optimization algorithms into the realm of deep learning, the more-than gigabytes large model parameters and gradients (Rajbhandari et al., 2019; Brown et al., 2020) have spurred interest in communication compression techniques to reduce the bandwidth requirements of training such models. While practical plug-and-play compressors already exist for communication in centralized deep learning (Seide et al., 2014; Vogels et al., 2019) that can retain full model quality at significant communication reductions, current compression algorithms in decentralized optimization require the tuning of additional hyperparameters. This is unfortunate, since running many experiments to tune these hyperparameters is especially challenging and costly in a decentralized environment.

In this paper, we study a specific class of low-rank compressors for decentralized optimization inspired by (Vogels et al., 2019; Cho et al., 2019) that are reliable and require no tuning. Like in their

work, we consider model parameters as matrices $\mathbf{X}$. Each pair of connected nodes $(i, j)$ repeatedly estimates the difference between their parameters $\mathbf{X}_i - \mathbf{X}_j$ through low-rank approximation. These approximations can be made without communicating full matrices due to the linearity of power iteration steps.

We validate these plug-and-play compressors on decentralized image classification and language modeling tasks, and show that we can achieve competitive performance to other methods that require additionally tuned hyperparameters. This allows users to tune a learning rate in a simpler centralized setup, and then transition to decentralized learning without extra effort. We prove hyperparameter-free convergence on a subclass of random low-rank approximations. For consensus, our method converges faster than prior methods (Koloskova et al., 2019b). For stochastic optimization, our rates are asymptotically independent of the compression rate.

## 2 Related work

**Communication compression in centrally coordinated learning.** Communication compression is an established approach to alleviate the communication bottleneck in parallel optimization in deep learning. While Alistarh et al. (2017); Wen et al. (2017); Seide et al. (2014); Bernstein et al. (2019); Karimireddy et al. (2019b) study gradient quantization, it is also possible to only send gradient coordinates with the largest absolute values Lin et al. (2018); Stich et al. (2018); Wangni et al. (2018).

It has become clear that linear compression operators are practical in the centralized setting because they enable efficient all-reduce aggregation (Yu et al., 2018; Vogels et al., 2019; Cho et al., 2019). Ivkin et al. (2019) use linear sketches to detect which parameter coordinates change most in a distributed setting. Wang et al. (2018) observed that gradients in deep learning can be well approximated as low-rank matrices.

The PowerSGD algorithm (Vogels et al., 2019), on which this work is based, is both linear and low-rank and performed well in a recent benchmark (Xu et al., 2020). An iteration of PowerSGD makes a low-rank approximation of the average error-corrected gradient across workers. The proposed decentralized scheme "PowerGossip" makes separate approximations for each pair of connected neighbors, directly approximating their pairwise model differences.

**Decentralized optimization.** Decentralized, or 'gossip'-based, optimization has been studied for many years (Tsitsiklis, 1984). Popular methods include those based on (stochastic) subgradient descent (Nedic & Ozdaglar, 2009) on node's local objective functions and with averaging between sparsely connected neighbors. Lian et al. (2017) evaluated the effectiveness of such schemes in the non-convex setting.

Tang et al. (2018) extend decentralized optimization with compressed communication, but require relatively high precision compression to ensure convergence. Koloskova et al. (2019a) and Tang et al. (2019) alleviate this constraint, supporting arbitrary-strength compression. Lu & Sa (2020) study a compression based on the assumption that model differences across connected nodes are coordinate-wise bounded. However, the abovementioned methods introduce additional hyperparameters specific to compression (e.g. the consensus stepsize)—an inconvenience we overcome in this work.

## 3 Decentralized machine learning

Decentralized multi-worker training of machine learning models has two key characteristics. Firstly, there is no central 'master' node and nodes can only communicate with a limited number of neighbors. This can either be a physical limitation of the network, or it can be desirable for performance. In a datacenter, sparse, decentralized connectivity leads to excellent scalability (Assran et al., 2019). The second characteristic is distributed data: each worker has their own data that potentially come from non-identical distributions. This can also be a hard limitation (e.g. to protect privacy), or it can be desirable for co-locality of computation and data.

The setup is formalized as follows: $n$ worker nodes aim to collectively minimize a loss function

$$f(\mathbf{X}) \coloneqq \frac{1}{n} \sum_{i=1}^{n} f_i(\mathbf{X}), \quad f_i(\mathbf{X}) \coloneqq \mathbb{E}_{\boldsymbol{\xi}_i \sim D_i} F_i(\mathbf{X}, \boldsymbol{\xi}_i)$$

over model parameters $\mathbf{X}$, where $f_i(\cdot)$ are smooth potentially non-convex loss functions over local data distributions $D_i$. We assume that $\mathbf{X} \in \mathbb{R}^{p \times q}$ where $p$ represents the size of the 'input' and $q$ is

the output size. For linear models, this matrix representation is natural. For multi-layer networks, each weight and bias is considered separately, and for convolutional layers, $q$ represents the number of input layers and the kernel size and $p$ is the number of output channels.

The network topology is represented by an undirected connected graph $G$ that connects nodes $i$ with their neighbors $\mathcal{N}_i$ (including self-links). Communication between nodes $i$ and $j$ is typically weighted by the $i, j$-th entry of a *mixing matrix* $\mathbf{W} \in \mathbb{R}^{n,n}$ which is non-zero only for connected nodes. This matrix is chosen such that for any scalars $\mathbf{v} \in \mathbb{R}^n$ held by the nodes, repeated averaging (gossip) between connected nodes, $\mathbf{W}\mathbf{v}$, gradually leads to consensus, $\mathbf{v}_i \to \frac{1}{n}\sum_{i=1}^n \mathbf{v}_i \ \forall i$.

In stochastic gradient-based optimization, each worker typically has its own model parameters $\mathbf{X}_i$. Gossip averaging is used to bring the $\mathbf{X}_i$'s closer together and share information between nodes, while local stochastic gradient updates change $\mathbf{X}_i$ to fit local data. Our methods builds on the elegant DP-SGD algorithm (Lian et al., 2017). In DP-SGD, for each timestep $t$ and each worker $i$,

$$\mathbf{X}_i^{(t+1)} := \mathbf{X}_i^{(t)} - \eta \nabla f_i(\mathbf{X}_i^{(t)}, \boldsymbol{\xi}_{i,t}) + \sum_{j \in \mathcal{N}_i} W_{ij}\left(\mathbf{X}_j^{(t)} - \mathbf{X}_i^{(t)}\right), \tag{1}$$

where $\eta_i$ is the learning rate and $\boldsymbol{\xi}_{i,t} \sim D_i$ represents a local data point. Note that each step requires sending and receiving the full model parameters between all pairs of connected neighbors, but that this communication can be overlapped with computation of the stochastic gradient.

## 4 Algorithm

Naively applying lossy communication compression (quantization / sparsification) to the gossip update in Eq. (1) leads to non-convergence. To support arbitrary compression, prior approaches introduce algorithmic modifications and additional hyperparameters to tune (Koloskova et al., 2019b; Tang et al., 2019, 2018). In this section, we introduce PowerGossip, a compressed consensus algorithm based on low-rank approximations and power iteration that does not suffer from these issues. Low-rank decomposition has already been shown to perform well in centralized deep learning (Vogels et al., 2019; Cho et al., 2019; Xu et al., 2020), and we find that they can be competitive with expensively tuned quantization- or sparsification-based algorithms for decentralized training as well.

PowerGossip is based on the premise that $\mathcal{C}_{\mathbf{v}}(\mathbf{X}) := (\mathbf{X}\mathbf{v})\mathbf{v}^\top$, for a matrix $\mathbf{X} \in \mathbb{R}^{p \times q}$ and vector $\mathbf{v} \in R^q$ with $\|\mathbf{v}\|_2 = 1$, can be a reasonable low-rank approximation of $\mathbf{X}$ that can be communicated with only $p$ floats instead of $p \times q$, given that all parties know $\mathbf{v}$. For the large weight matrices in deep learning, this reduction is significant. For a random $\mathbf{v}$, $\mathcal{C}_{\mathbf{v}}$ is a random projection, while for $\mathbf{v}$ being the top right singular vector, $\mathcal{C}_{\mathbf{v}}(\mathbf{X})$ is the best rank-1 approximation of $\mathbf{X}$ in the Frobenius norm.

We use the low-rank compressor $\mathcal{C}_{\mathbf{v}}$ to reduce communication in the gossip part of Eq. (1):

$$\mathbf{X}_i^{(t+1)} := \mathbf{X}_i^{(t)} + \sum_{j \in \mathcal{N}_i} W_{ij}\,\mathcal{C}_{\mathbf{v}_{ij}}(\mathbf{X}_j^{(t)} - \mathbf{X}_i^{(t)}), \tag{2}$$

for a time-varying vector $\mathbf{v}_{ij}$ shared between each pair of connected workers. Due to linearity, $\mathcal{C}_{\mathbf{v}}(\mathbf{X}_j - \mathbf{X}_i) = (\mathbf{X}_j - \mathbf{X}_i)\mathbf{v}\mathbf{v}^\top = (\mathbf{X}_j\mathbf{v} - \mathbf{X}_i\mathbf{v})\mathbf{v}^\top$. Therefore, the compressed difference can be computed jointly by nodes $i$ and $j$ without ever communicating the full $\mathbf{X}_j - \mathbf{X}_i$. Thus any nodes $i$ and $j$ only need to exchange vectors instead of matrices.

The approximation quality of $\mathcal{C}_{\mathbf{v}}$ depends on the choice of the projection vector $\mathbf{v}$, and we leverage the mechanism of power iteration to find good ones. Every time $(k)$ the compressor $\mathcal{C}_{\mathbf{v}}$ is used on some parameter difference $\mathbf{D}^{(k)} := \mathbf{X}_j^{(k)} - \mathbf{X}_i^{(k)}$, we choose $\mathbf{v}^{(k)}$ based on the previous low-rank approximation. Starting with a random initial vector $\mathbf{v}^{(0)}$, we use

$$\mathbf{v}^{(2k+1)} := \frac{\mathbf{D}^{(2k)}\mathbf{v}^{(2k)}}{\|\mathbf{D}^{(2k)}\mathbf{v}^{(2k)}\|}, \qquad \mathbf{v}^{(2k)} := \frac{\mathbf{D}^{(2k-1)\top}\mathbf{v}^{(2k-1)}}{\|\mathbf{D}^{(2k-1)\top}\mathbf{v}^{(2k-1)}\|}, \qquad \forall k \in \mathbb{Z}_{\geq 0}. \tag{3}$$

If $\mathbf{X}_j^{(k)} - \mathbf{X}_i^{(k)}$ changes slowly over time, this procedure approaches power iteration and it finds the top eigenvector $\mathbf{v}$. This approach empirically leads to better approximations and faster convergence than compression with random projections.

Algorithm 1 describes how we use PowerGossip for stochastic optimization. Algorithm 2 presents the details of our compression scheme.

---

**Algorithm 1** Decentralized SGD with edge-wise compression

---
1: **input** model parameters $\mathbf{X}_i^{(0)} \in \mathbb{R}^{p \times q}$ for each node $i$ out of $n$, randomly initialized identically
2: **given** a symmetric, doubly stochastic, diffusion matrix $\mathbf{W} \in \mathbb{R}^{N \times N}$
3: **given** a compressor $\mathcal{C}$ that can approximate $\mathbf{X}_i - \mathbf{X}_j$ with little communication
4: **for** each timestep $t$ **at** each worker $i$ **do**
5: $\quad\quad \mathbf{G} \leftarrow$ a stochastic gradient $\nabla f(\mathbf{X}_i^{(t-1)}, \boldsymbol{\xi}_{i,t})$ for mini-batch $\boldsymbol{\xi}_{i,t}$
6: $\quad\quad \mathbf{X}_i^{(t)} \leftarrow \mathbf{X}_i^{(t-1)} + \sum_{j \in \mathcal{N}_i} W_{ij} \mathcal{C}(\mathbf{X}_j^{(t-1)} - \mathbf{X}_i^{(t-1)}) - \eta \cdot \mathbf{G}$
7: **end for**

---

---

**Algorithm 2** Rank-1 $s$-step PowerGossip compression for Algorithm 1

---
1: **initialize** a projection vector $\mathbf{v}_{ij} = -\mathbf{v}_{ji} \in \mathbb{R}^q$ for each pair of connected nodes $i, j$, initialized from an entry-wise standard normal distribution, stored on nodes $i$ and $j$. Initialize $k \leftarrow 0$.
2: **procedure** $\mathcal{C}(\mathbf{X}_j - \mathbf{X}_i)$
3: $\quad$ **for** $s$ power iteration steps **do**
4: $\quad\quad$ **increment** $k \leftarrow k + 1$
5: $\quad\quad$ **if** $k \equiv 1 \bmod 2$ **then**
6: $\quad\quad\quad \hat{\mathbf{v}} \leftarrow \frac{\mathbf{v}_{ij}}{\|\mathbf{v}_{ij}\|}$
7: $\quad\quad\quad \mathbf{p}_j \leftarrow \mathbf{X}_j \hat{\mathbf{v}}, \quad\quad \mathbf{p}_i \leftarrow \mathbf{X}_i \hat{\mathbf{v}}$ $\quad\quad\quad\quad\quad\quad\quad$ ▷ computed on nodes $i$ and $j$
8: $\quad\quad\quad \hat{\mathbf{Q}} \leftarrow (\mathbf{p}_j - \mathbf{p}_i) \hat{\mathbf{v}}^\top$
9: $\quad\quad\quad \mathbf{v}_{ij} \leftarrow \mathbf{p}_j - \mathbf{p}_i$ $\quad\quad\quad\quad\quad\quad\quad\quad\quad$ ▷ $\mathbf{v}_{ij}$ changes between $\mathbb{R}^p$ and $\mathbb{R}^q$
10: $\quad\quad$ **else**
11: $\quad\quad\quad$ do the same, but with $\mathbf{X}$ transposed as in Eq. (3).
12: $\quad\quad$ **end if**
13: $\quad$ **end for**
14: $\quad$ **return** the approximation $\hat{\mathbf{Q}}$
15: **end procedure**
16: **note** that computations of $\mathcal{C}(\mathbf{X}_j - \mathbf{X}_i) = -\mathcal{C}(\mathbf{X}_i - \mathbf{X}_j)$ overlap and share communication.

---

## 4.1 Properties

**Linearity.** Due to the linearity of matrix multiplication, we can compute a matrix-vector product $(\mathbf{X}_i - \mathbf{X}_j)\mathbf{v}$ with matrices stored on different workers in a distributed fashion as $(\mathbf{X}_i \mathbf{v}) - (\mathbf{X}_j \mathbf{v})$. This circumvents communication of matrices by sending much smaller vectors instead. By compressing the differences of the models, we ensure that the models get closer to the average in every step without the need for additional 'consensus stepsize' like prior protocols. In particular, if two workers agree on the parameters and their difference is 0, then the compressed update will also be 0. This ensures that consensus is always a fixed-point of our method for arbitrary-strength compressors.

**Low-rank compression.** PowerGossip approximates differences between model parameters by low-rank matrices. The quality of these approximations depends on the power spectra of the differences. Similar to how top-$k$ compression—which approximates a vector by its top $k$ coordinate in absolute value, and zeros otherwise—works best when a few coordinates are much larger than the rest, low-rank compression can leverage the peaky power spectra found in deep learning (Vogels et al., 2019; Cho et al., 2019) to maximize information sent per bit. Our experiments in Section 6 confirm that low-rank compression is competitive with quantization- or sparsification-based approaches, while keeping our algorithm simple and free of hyperparameters.

**Memory and computation complexity.** The linear projection operations in PowerGossip are well suited for accelerator hardware used in deep learning (Vogels et al., 2019; Cho et al., 2019; Xu et al., 2020), and are typically even faster than compression based on random sparsification or quantization. Like in DP-SGD (Lian et al., 2017), this computation and the communication between nodes can be overlapped with gradient computation. Storing the previous projection vectors $\mathbf{v}$ requires memory linear in the number of connections per worker, but these vectors are very small compared to a full model (0.1–2% of the full model in our experiments). This yields lower memory usage than competing methods ChocoGossip Koloskova et al. (2019a) and DeepSqueeze Tang et al. (2019).

# 5 Theoretical analysis

## 5.1 Assumptions and setup

**Loss functions.** We make standard assumptions about our loss functions. Note that our analysis covers both functions satisfying (A1), as well as more general non-convex functions which do not.

**(A1)** $f_i$ is $\mu$-**convex** for $\mu \geq 0$ if it satisfies for any $\mathbf{X}$, and $\mathbf{X}^\star$ minimizing $f$

$$\nabla f_i(\mathbf{X}) \circ (\mathbf{X}^\star - \mathbf{X}) \leq -\Big(f_i(\mathbf{X}) - f_i(\mathbf{X}^\star) + \frac{\mu}{2}\|\mathbf{X} - \mathbf{X}^\star\|_F^2\Big).$$

**(A2)** We assume $\{f_i\}$ are $L$-**smooth** and thus satisfy:
$$\|\nabla f_i(\mathbf{X}) - \nabla f_i(\mathbf{Y})\|_F \leq L\|\mathbf{X} - \mathbf{Y}\|_F, \text{ for any } i, \mathbf{X}, \mathbf{Y}.$$

**(A3)** **Bounded variance:** We assume there exist constants $\sigma^2$ and $\zeta^2$ which bound the variance within and across different nodes, i.e. for any $\mathbf{X}$ we have

$$\mathbb{E}_{\boldsymbol{\xi}_i \sim D_i}\|\nabla F_i(\mathbf{X}, \boldsymbol{\xi}_i) - \nabla f_i(\mathbf{X})\|_F^2 \leq \sigma^2 \quad \text{and} \quad \tfrac{1}{N}\sum_{i=1}^N \|\nabla f_i(\mathbf{X}) - \nabla f(\mathbf{X})\|_F^2 \leq \zeta^2.$$

Assumption A1 is known as *star-convexity* and is weaker than the usual definition of convexity (Stich & Karimireddy, 2019). While A3 requires both the variance within each node as well as across the nodes be bounded, we allow for heterogeneous (non-iid) data distributions across the nodes.

**Communication network.** We assume that we are given a mixing matrix $\mathbf{W} \in \mathbb{R}^{n \times n}$ and an underlying communication network over $n$ nodes $([n], E)$ satisfying (A4):

**(A4)** $W_{ij} \neq 0$ only if $(i,j) \in E$, and $\mathbf{W} \in \mathbb{R}^{n \times n}$ is symmetric $(\mathbf{W}^\top = \mathbf{W})$ and doubly stochastic $(\mathbf{W}\mathbf{1} = \mathbf{1}, \mathbf{1}^\top\mathbf{W} = \mathbf{1}^\top)$. Further, $\mathbf{W}^2$ has eigenvalues $1 = \lambda_1^2 \geq \lambda_w^2 \geq \ldots \lambda_n^2$ with **spectral gap** $\rho := 1 - \lambda_2^2 > 0$.

Assumption (A4) characterizes the mixing matrix $\mathbf{W}$ for decentralized optimization and controls the rate of information spread in the network (Lian et al., 2017; Pu & Nedic, 2018). If $\mathbf{W}$ satisfies (A4) for $\rho > 0$, then the underlying communication network is undirected and strongly connected.

**Compression operators.** We introduce a new class of compression operators $\mathcal{C}(\cdot)$ and assume that every compressor used in Algorithm 1 satisfies (A5):

**(A5)** We assume that $\mathcal{C}$ is a $\delta$-approximate unbiased **linear projection** operator i.e. for any $\mathbf{X}$ and $\mathbf{Y}$, the following are true for some $\delta > 0$:
$$\mathcal{C}(\mathbf{X} + \mathbf{Y}) = \mathcal{C}(\mathbf{X}) + \mathcal{C}(\mathbf{Y}), \quad \mathcal{C}(\mathcal{C}(\mathbf{X})) = \mathcal{C}(\mathbf{X}), \quad \text{and} \quad \mathbb{E}[\mathcal{C}(\mathbf{X})] = \delta X.$$

Consider a random-$p$ sampler whose $(i,j)$ element $[\mathcal{S}_p(\mathbf{X})]_{i,j}$ is $X_{i,j}$ with probability $p$ and 0 otherwise. Then $\mathcal{S}_p(\cdot)$ is a linear projection operator satisfying (A5) with $\delta = p$.

For a second example closer to Algorithm 2, consider the following compressor for $\mathbf{X} \in \mathbb{R}^{p,q}$:

$$\mathcal{R}(\mathbf{X}) := (\mathbf{X}\mathbf{u})\mathbf{u}^\top \text{ for } \mathbf{u} \sim S^{(q-1)},$$

i.e. we project $\mathbf{X}$ along $\mathbf{u}$ which is sampled uniformly from the unit sphere. The operator $\mathcal{R}(\mathbf{X})$ approximates $\mathbf{X}$ as a product of two rank-1 matrices $\mathbf{u}$ and $\mathbf{Xu}$. Then, $\mathcal{R}(\cdot)$ is clearly linear in $\mathbf{X}$, is an unbiased projection operator, and satisfies (A5) with $\delta = \frac{1}{q}$. We can also approximate $\mathbf{X}$ by two rank-$k$ matrices as $\mathcal{R}_k(\mathbf{X}) = (\mathbf{X}\mathbf{U})\mathbf{U}^\top$ for $\mathbf{U} \in \mathbb{R}^{q \times k}$ being a uniformly sampled orthonormal matrix. Then $\mathcal{R}_k(\cdot)$ satisfies (A5) with $\delta = \frac{k}{q}$. We can also define a left projection operator $\mathcal{L}(\mathbf{X}) := \mathbf{v}(\mathbf{v}^\top\mathbf{X})$ for $\mathbf{v} \sim S^{(p-1)}$. The operator $\mathcal{L}(\cdot)$ approximates $\mathbf{X}$ with two rank-1 matrices $\mathbf{v}$ and $\mathbf{X}^\top\mathbf{v}$ and satisfies (A5) with $\delta = \frac{1}{p}$.

While (A5) defines a specific class of compression operators which are a subset of those considered in (Koloskova et al., 2019b), they can still be of arbitrary approximation quality $\delta > 0$.

## 5.2 Convergence rates

We study the rate of consensus as well as convergence of the objective function in stochastic optimization with compressed communication. Our analysis shows that our algorithm is not only simpler than the previous approaches, but also significantly faster. To simplify notation, we will use $\bar{\cdot}$ to indicate the average across the $n$ nodes, e.g. $\bar{\mathbf{X}} := \frac{1}{n}\sum_{i=1}^n \mathbf{X}_i$.

**Compressed consensus.** Suppose that every iteration, each worker $i$ performs the following update:

$$\mathbf{X}_i^{(t)} := \mathbf{X}_i^{(t-1)} + \sum_{j \in \mathcal{N}_i} W_{ij}\left(\mathcal{C}_{ij}^{(t)}(\mathbf{X}_j^{(t-1)}) - \mathcal{C}_{ij}^{(t)}(\mathbf{X}_i^{(t-1)})\right). \tag{4}$$

Each edge $(i,j)$ can use a different compressor $\mathcal{C}_{ij}^{(t)}$ that can be varied over time. In this update, only compressed parameters are communicated.

**Theorem I.** *Assuming all compressors $\mathcal{C}_{ij}^{(t)}$ are $\delta$-approximate satisfying (A5) and that the mixing matrix $\mathbf{W}$ has spectral gap $\rho$ as in (A4), then the update (4) achieves consensus at a $q$-linear rate:*

$$\frac{1}{N}\sum_{i=1}^{N} \mathbb{E}\left\|\mathbf{X}_i^{(t)} - \bar{\mathbf{X}}^{(0)}\right\|_F^2 \le (1-\rho\delta)\frac{1}{N}\sum_{i=1}^{N}\left\|\mathbf{X}_i^{(t-1)} - \bar{\mathbf{X}}^{(0)}\right\|_F^2.$$

Note that update (4) requires no additional parameters and that our rate is linear in both $\delta$ and $\rho$. When $\delta = 1$, i.e. with uncompressed messages, the rate in I corresponds to the classical consensus rate (e.g. Xiao & Boyd, 2004a). In contrast, (Koloskova et al., 2019b) require a consensus stepsize, do not obtain $q$-linear rates, and are slower with a rate depending on $\rho^2\delta$ instead of our $\rho\delta$.

**Compressed optimization.** Consider the following algorithm where every node $i$ performs the following updates using a sequence of predetermined stepsizes $\{\eta_t\}$:

$$\begin{aligned}
\mathbf{Y}_i^{(t)} &:= \mathbf{X}_i^{(t-1)} - \eta_t \nabla F_i(\mathbf{X}, \boldsymbol{\xi}_{i,t}) \\
\mathbf{X}_i^{(t)} &:= \mathbf{Y}_i^{(t)} + \sum_{j \in \mathcal{N}_i} W_{ij}(\mathcal{C}_{ij}^{(t)}(\mathbf{Y}_j^{(t)}) - \mathcal{C}_{ij}^{(t)}(\mathbf{Y}_i^{(t)})).
\end{aligned} \tag{5}$$

This algorithm is like PowerGossip, but it applies the consensus update of (4) after a local gradient update rather than simultaneously. Again, the compressors are allowed to vary across edges and with time, and only compressed parameters are communicated. After running for $T$ steps, we will randomly pick the final model given some weights $\{\alpha_t\}$ as

$$\mathbf{X}_i^{\text{out}} := \mathbf{X}_i^{(t)} \text{ with probability proportional to } \alpha_t. \tag{6}$$

**Theorem II.** *Suppose that assumptions A2–A5 hold at every round of (5). Then, in each of the following cases there exist a sequence of stepsizes $\{\eta_t\}$ and weights $\{\alpha_t\}$ such that the output $\bar{\mathbf{X}}^{out}$ computed using (5) and (6) is $\varepsilon$-accurate.*

- **Non-convex:** $\mathbb{E}\|\nabla f(\bar{\mathbf{X}}^{out})\|^2 \le \varepsilon$ *after*
$$T = \mathcal{O}\left(\frac{L\sigma^2}{n\varepsilon^2} + \frac{\sqrt{L}(\zeta+\sigma)}{\rho\delta\varepsilon^{3/2}} + \frac{L}{\rho\delta\varepsilon}\right) \text{ rounds.}$$

- **Convex:** *If $\{f_i\}$ are convex and satisfy (A1) with $\mu = 0$, then $\mathbb{E}[f(\bar{\mathbf{X}}^{out})] \le \varepsilon$ after*
$$T = \mathcal{O}\left(\frac{\sigma^2}{n\varepsilon^2} + \frac{\zeta+\sigma}{\rho\delta\varepsilon^{3/2}} + \frac{L}{\rho\delta\varepsilon}\right) \text{ rounds.}$$

- **Strongly-convex:** *If $\{f_i\}$ satisfy (A1) with $\mu > 0$, then $\mathbb{E}[f(\bar{\mathbf{X}}^{out})] \le \varepsilon$ after*
$$T = \tilde{\mathcal{O}}\left(\frac{\sigma^2}{n\mu\varepsilon} + \frac{\zeta+\sigma}{\rho\delta\mu\sqrt{\varepsilon}} + \frac{L}{\rho\delta\mu}\log\left(\frac{1}{\varepsilon}\right)\right) \text{ rounds.}$$

Let us focus on the strongly convex case ignoring logarithmic factors. Theorem II proves that the iteration complexity is $\frac{\sigma^2}{n\mu\varepsilon} + \frac{\zeta+\sigma}{\rho\delta\mu\sqrt{\varepsilon}} + \frac{L}{\rho\delta\mu}\log\left(\frac{1}{\varepsilon}\right)$. This can be decomposed into three terms. The first stochastic term $\frac{\sigma^2}{n\mu\varepsilon}$ is independent of both the compression factor $\delta$ as well as spectral-gap $\rho$ implying that these terms do not affect the asymptotic rates. It scales linearly with the number of nodes $n$. The second term $\frac{\zeta+\sigma}{\rho\delta\mu\sqrt{\varepsilon}}$ corresponds to the *drift* experienced and is a penalty due to computation of gradients at inexact points (Karimireddy et al., 2019a). However, this is asymptotically smaller than the stochastic term. Last is the optimization term $\frac{L}{\rho\delta\mu}\log\left(\frac{1}{\varepsilon}\right)$, which is the slowed down by a factor of $\rho\delta$. If $\rho\delta = 1$, this term matches the linear rate of gradient descent on strongly convex functions (Nesterov, 2004). In contrast, the optimization term of (Koloskova et al., 2019b) is sub-linear. The dependence on $\rho$ and $\delta$ is linear in our rates while (Koloskova et al., 2019b) have a quadratic dependence on $\rho$. With exact communication ($\delta = 1$) we recover the rates of (Koloskova et al., 2020).

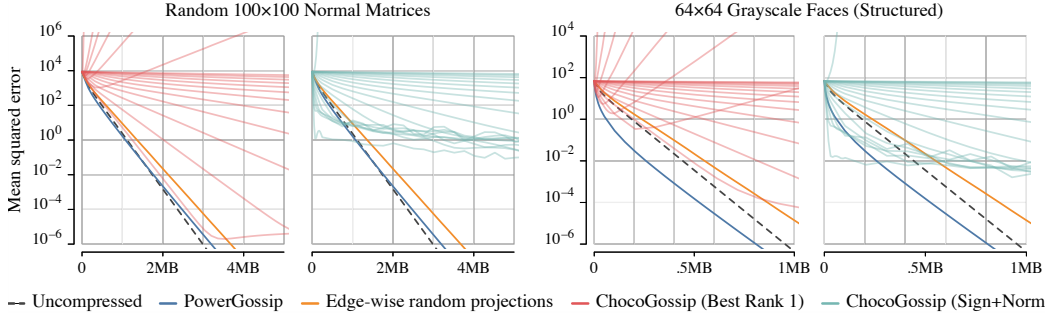

Figure 1: Consensus in an 8-ring. We study the level of consensus achieved as a function of bits transmitted by decentralized averaging. We compare out-of-the-box PowerGossip with power iterations and random projections against ChocoGossip (Koloskova et al., 2019b) with varying diffusion parameters. PowerGossip is competitive to the best tuned instances of ChocoGossip, and can leverage low rank structure in structured data (right).

## 6 Experimental analysis

We study PowerGossip in three settings. We first evaluate bits of communication required to reach **consensus** between 8 workers in a ring through (compressed) gossip averaging. The workers start with personal data matrices $\mathbf{X}_i$ ($i = 1 \ldots 8$) that are either *unstructured*, from a $100 \times 100$ standard normal distribution, or *structured*, with $64 \times 64$ images from the Faces Database (AT&T Laboratories Cambridge). Then we evaluate PowerGossip in deep learning. We study the algorithm on the Cifar-10 **image classification** benchmark of Koloskova et al. (2019a), using a ResNet-20 and labeled images that are reshuffled between 8 workers every epoch. We also follow the **language modeling** experiment on WikiText-2 with an LSTM from Vogels et al. (2019) and extend it to a decentralized setting with 16 workers in a ring. Here, the training data is strictly partitioned between workers, dividing the source text equally over the workers in the original ordering.

In all experiments, we tune the hyperparameters of our baselines according to Appendix G and use the same learning rate as uncompressed centralized SGD for all instances of PowerGossip. Further details on the experimental settings are specified in Appendix C.

**Random projections v.s. power iteration.** Power iteration helps PowerGossip to leverage approximate low-rank structure in parameter differences between workers. This is illustrated by the consensus experiments in Figure 1. While on random data no compressed gossip algorithm outperforms full-precision gossip in bits to an arbitrary level of consensus, PowerGossip can reliably use structure in images of faces AT&T Laboratories Cambridge with less communication.

In our deep learning experiments, we also observe that PowerGossip requires less communication than random projections. The table on the right shows that more efficient communication leads to improved test accuracy within a fixed budget of 90 epochs.

| Algorithm | | Test loss |
|---|---|---|
| PowerGossip | w/ Random projections | 4.627 |
| | w/ Power iteration | 4.565 |
| DP-SGD | 35× communication | 4.583 |

**Compression rate.** The compression rate in PowerGossip is determined by the number of power iteration steps per stochastic gradient update. For models with large, square parameter tensors, like our LSTM (Appendix I), a single step of PowerGossip uses less than $0.1\%$ of the bits used by an uncompressed averaging step. For a smaller model like the ResNet-20, the compression ratio is much lower. While our algorithm works for any compression rate, more gradient steps may be required to reach the same accuracy under extreme compression.

In our experiments, we use compression levels similar to those studied in related work. At those levels, PowerGossip achieves test performance similar to uncompressed DP-SGD in the same number of steps. Our compression level is varied through the number of power iterations per gradient update. More power iteration steps speed up consensus at the cost of increased communication in the same way as increasing the rank of the compressor does (see Appendix F), but it requires less memory to store the previous approximation and avoids an expensive orthogonalization step (Vogels et al., 2019). Table 1 shows the effect of varying our compression rate while keeping the number of epochs fixed.

| Algorithm | $\eta$ | $\gamma$ | Test loss | Sent/epoch |
|---|---|---|---|---|
| All-reduce (baseline) | tuned | | 4.46 | |
| Uncompressed (DP-SGD) | tuned | | 4.58 | 15.0 GB |
| PowerGossip (8 iterations) | default | | 4.73 | 127 MB (122×) |
| PowerGossip (16 iterations) | default | | 4.63 | 230 MB (67×) |
| PowerGossip (32 iterations) | default | | 4.57 | 437 MB (35×) |
| Choco (Sign+Norm) | tuned | tuned | 4.49 | 483 MB (32×) |
| Choco (top-1%) | tuned | tuned | 5.04 | 464 MB (33×) |

Table 1: Test loss achieved within 90 epochs on WikiText-2 language modeling with an LSTM on a 16-ring with strictly partitioned training data. PowerGossip requires no tuning, supports varying levels of compression, and is competitive to tuned ChocoSGD (Koloskova et al., 2019a) at a similar compression rate, matching the test loss of uncompressed DP-SGD.

| Algorithm | $\eta$ | $\gamma$ | $\theta$ | Test accuracy | Sent/epoch |
|---|---|---|---|---|---|
| All-reduce (baseline) | tuned | | | 92.3% | |
| Uncompressed (DP-SGD) | tuned | | | 92.1% | 102 MB |
| Choco (top-1%) | tuned | tuned | | 91.2% | 3.1 MB (33×) |
| Choco (Sign+Norm) | tuned | tuned | | 92.0% | 3.2 MB (32×) |
| Moniqua (2-bit) | tuned | tuned | tuned | 90.7% | 6.4 MB (16×) |
| DeepSqueeze (Sign+Norm) | tuned | tuned | | 91.2% | 3.2 MB (32×) |
| PowerGossip (1 iteration) | default | | | 91.7% | 1.8 MB (57×) |
| PowerGossip (2 iterations) | default | | | 91.9% | 3.0 MB (34×) |

Table 2: Test accuracy reached on Cifar-10 within 300 epochs with a ResNet-20 by decentralized optimization algorithms. PowerGossip has no additional hyperparameters and is competitive to all related work at a similar compression rate. Other algorithms used tuned learning rate $\eta$, averaging stepsize $\gamma$. Moniqua has an additional parameter $\theta$ that can be computed or tuned.

**Hyper-parameter tuning.** In our experiments, we have strictly used the same learning rate tuned for centralized, uncompressed SGD for all PowerGossip configurations. Tables 1 and 2 show that we can reach performance competitive to DP-SGD in both tasks, at a similar compression rate to the best tuned configurations of ChocoSGD (Koloskova et al., 2019b) and DeepSqueeze (Tang et al., 2019).

# 7 Conclusion

The introduction of communication compression to decentralized learning has come with algorithmic changes that introduced new hyperparameters required to support arbitrary compression operators. Focusing on a special class of linear low-rank compression, we presented simple parameter-free algorithms that perform as well as the extensively tuned alternatives in decentralized learning. Using power-iterations, this method can leverage the approximate low-rank structure present in deep learning updates to maximize the information transferred per bit, and reduce the communication between workers significantly at no loss in quality compared to full-precision decentralized algorithms. This is achieved with lower memory consumption than current state-of-the-art decentralized optimization algorithms that use communication compression.

Plug-and-play algorithms like PowerGossip can be directly deployed in a decentralized setting while reusing standard learning rates set in the centralized environment without compression. In view of the environmental, financial, and productivity impact of hyperparameter tuning in deep learning, such tuning-free methods are crucial for practical applicability of communication compression in decentralized machine learning.

## 8 Broader Impact

We believe that the field of decentralized learning plays a key role in translating the recent successes in deep learning from large organizations with large centralized datasets to smaller industry players and individuals. In particular, decentralized and therefore collaborative training on decentralized data is an important building block towards helping to better align each individual's data ownership and privacy with the resulting utility from jointly trained machine learning models. The ability to train collaboratively on decentralized data may lead to transformative insights in many fields, especially in applications where data is user-provided and privacy sensitive (Nedic, 2020). In addition to privacy, efficiency gains in distributed training reduce the environmental impact of training large machine learning models. The introduction of a practical and reliable communication compression technique is a small step towards achieving these goals on collaborative privacy-preserving and efficient decentralized learning.

## 9 Funding Disclosure

This project was supported by SNSF grant 200021_175796, as well as a Google Focused Research Award. The experiments were run using Google Cloud credits donated by Google.

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
