[Supplementary Material]

# Practical Low-Rank Communication Compression in Decentralized Deep Learning

## Abstract

Lossy gradient compression has become a practical tool to overcome the communication bottleneck in centrally coordinated distributed training of machine learning models. However, algorithms for decentralized training with compressed communication over arbitrary connected networks have been more complicated, requiring additional memory and hyperparameters. We introduce a simple algorithm that directly compresses the model differences between neighboring workers using low-rank linear compressors applied to model differences. Inspired by the PowerSGD algorithm for centralized deep learning (Vogels et al., 2019), this algorithm uses power iteration steps to maximize the information transferred per bit. We prove that our method requires no additional hyperparameters, converges faster than prior methods, and is asymptotically independent of both the network and the compression. Out of the box, these compressors perform on par with state-of-the-art tuned compression algorithms in a series of deep learning benchmarks.

## 1 Introduction

The major advances in machine learning in the last decade have been made possible by very large datasets collected by multifaceted organizations. We live in a society where almost every individual owns electronic devices that collect huge amounts of data, which—when used collaboratively—could lead to transformative insights (Nedic, 2020). Often this data is bound to the device it is captured on. This might be for practical reasons of efficiency, or for more fundamental reasons such as privacy constraints. Centralized systems present a single point of failure both for data transfer, as well as for information security and privacy (Kairouz et al., 2019).

The paradigm of decentralized machine learning is key to leveraging the potential of this new kind of data. In this model, each connected device (node) has its own data. Each node can only communicate with few others, and together, the network of sparsely connected nodes aims to collaboratively train a model that minimizes a loss function on their joint dataset. The decentralized approach is not only useful in fundamentally decentralized systems, but the sparse communication patterns can sometimes even lead to efficiency gains in datacenter settings (Assran et al., 2019).

In bringing decentralized optimization algorithms into the realm of deep learning, the more-than gigabytes large model parameters and gradients (Rajbhandari et al., 2019; Brown et al., 2020) have spurred interest in communication compression techniques to reduce the bandwidth requirements of training such models. While practical plug-and-play compressors already exist for communication in centralized deep learning (Seide et al., 2014; Vogels et al., 2019) that can retain full model quality at significant communication reductions, current compression algorithms in decentralized optimization require the tuning of additional hyperparameters. This is unfortunate, since running many experiments to tune these hyperparameters is especially challenging and costly in a decentralized environment.

In this paper, we study a specific class of low-rank compressors for decentralized optimization inspired by (Vogels et al., 2019) that are reliable and require no tuning. Our low-rank compressor considers model parameters as matrices $\mathbf{X}$, and runs power iterations on the difference of two node's parameters $\mathbf{X}_i - \mathbf{X}_j$ to obtain a good low-rank approximation. Because these steps are linear, they can be executed in a distributed fashion, avoiding the expensive communication of full matrices.

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

## A  Compressed Consensus (Proof of Theorem I)

Recall that the consensus update for each node $i$ performs (4):

$$\mathbf{X}_i^{(t)} = \mathbf{X}_i^{(t-1)} + \sum_{j \in \mathcal{N}_i} W_{ij}(\mathcal{C}_{ijt}(\mathbf{X}_j^{(t-1)}) - \mathcal{C}_{ijt}(\mathbf{X}_i^{(t-1)})) \,.$$

**Lemma 1** (Preserves average)**.** *For every step of* (4)*, $\bar{\mathbf{X}}^{(t)} = \bar{\mathbf{X}}^{(0)}$.*

*Proof.* Note that for every edge $(i,j) \in E$, we add to node $i$ exactly what is subtracted from node $j$. This preserves the average:

$$
\begin{aligned}
\bar{\mathbf{X}}^{(t)} &= \frac{1}{n} \sum_{i=1}^{n} \left( \mathbf{X}_i^{(t-1)} + \sum_{j \in \mathcal{N}_i} W_{ij}(\mathcal{C}_{ijt}(\mathbf{X}_j^{(t-1)}) - \mathcal{C}_{ijt}(\mathbf{X}_i^{(t-1)})) \right) \\
&= \bar{\mathbf{X}}^{(t-1)} + \frac{1}{n} \sum_{(i,j) \in E} \left( W_{ij}(\mathcal{C}_{ijt}(\mathbf{X}_j^{(t-1)}) - \mathcal{C}_{ijt}(\mathbf{X}_i^{(t-1)})) + W_{ji}(\mathcal{C}_{jit}(\mathbf{X}_i^{(t-1)}) - \mathcal{C}_{jit}(\mathbf{X}_j^{(t-1)})) \right) \\
&= \bar{\mathbf{X}}^{(t-1)} \,.
\end{aligned}
$$

The last equality follows because $W_{ij} = W_{ji}$ and $\mathcal{C}_{ijt} = \mathcal{C}_{jit}$. □

**Lemma 2** (Effect of compression)**.** *Assuming* (A4) *and* (A5) *hold, the iteration* (4) *satisfies*

$$\|\boldsymbol{\Delta}_i^{(t)}\|_F^2 \leq (1-\delta)\|\boldsymbol{\Delta}_i^{(t-1)}\|_F^2 + \delta\|\sum_{j \in [N]} W_{ij}\boldsymbol{\Delta}_j^{(t-1)}\|_F^2 \,.$$

*where we define $\boldsymbol{\Delta}_i^{(t)} := \mathbf{X}_i^{(t)} - \bar{\mathbf{X}}^{(0)}$.*

*Proof.* Starting from the consensus update and the fact that $\sum_j W_{ij} = 1$, we have

$$
\begin{aligned}
\mathbf{X}_i^{(t)} &= \mathbf{X}_i^{(t-1)} + \sum_{j \in \mathcal{N}_i} W_{ij}(\mathcal{C}_{ijt}(\mathbf{X}_j^{(t-1)}) - \mathcal{C}_{ijt}(\mathbf{X}_i^{(t-1)})) \\
&= \mathbf{X}_i^{(t-1)} + \sum_{j \in [N]} W_{ij}(\mathcal{C}_{ijt}(\mathbf{X}_j^{(t-1)} - \mathbf{X}_i^{(t-1)})) \\
&= \mathbf{X}_i^{(t-1)} + \sum_{j \in [N]} W_{ij}\Pi_{ijt}(\mathbf{X}_j^{(t-1)} - \mathbf{X}_i^{(t-1)}) \,.
\end{aligned}
$$

The second equality used $W_{ij} \neq 0$ only if $(i,j) \in E$ and the linearity of the compressor. Finally, since $\mathcal{C}_{ijt}$ is a linear projection, we can replace it by a projection matrix $\Pi_{ijt}$. Recall that $\mathcal{C}_{ijt}$ is an $\delta$-approximate linear projection which implies that $\Pi_{ijt}$ satisfies

$$\mathbb{E}[\Pi_{ijt}] = \mathbb{E}[\Pi_{ijt}^\top] = \mathbb{E}[\Pi_{ijt}^\top \Pi_{ijt}] = \delta I \,. \tag{7}$$

Further, since $\Pi_{ijt}$ is a projection matrix, we have for any $i,j$

$$
\begin{aligned}
&\Pi_{ijt}^\top \preceq I \\
\Rightarrow &\Pi_{ijt}^\top \Pi_{ikt} \preceq \Pi_{ikt} \\
\Rightarrow &\mathbb{E}[\Pi_{ijt}^\top \Pi_{ikt}] \preceq \mathbb{E}[\Pi_{ikt}] = \delta I \,.
\end{aligned}
$$

Note that we did not require any sort of independence between the projections $\Pi_{ijt}^\top \Pi_{ikt}$ in the above derivation. Armed with these properties of the projection matrices, we turn our attention to the error term defined as $\boldsymbol{\Delta}_i^{(t)} := \mathbf{X}_i^{(t)} - \bar{\mathbf{X}}^{(0)}$. Our previous expression for $\mathbf{X}_i^{(t)}$ implies that

$$\boldsymbol{\Delta}_i^{(t)} = \boldsymbol{\Delta}_i^{(t-1)} + \sum_{j \in [n]} W_{ij}\Pi_{ijt}(\boldsymbol{\Delta}_j^{(t-1)} - \boldsymbol{\Delta}_i^{(t-1)}) \,.$$

Expanding $\boldsymbol{\Delta}_i^{(t)^\top}\boldsymbol{\Delta}_i^{(t)}$ and taking expectations on both sides gives

$$\mathbb{E}[\boldsymbol{\Delta}_i^{(t)^\top}\boldsymbol{\Delta}_i^{(t)}] = \boldsymbol{\Delta}_i^{(t-1)^\top}\boldsymbol{\Delta}_i^{(t-1)} + \sum_{j\in[n]} W_{ij}\boldsymbol{\Delta}_i^{(t-1)^\top}\mathbb{E}[\Pi_{ijt}](\boldsymbol{\Delta}_j^{(t-1)} - \boldsymbol{\Delta}_i^{(t-1)})$$

$$+ \sum_{j\in[n]} W_{ij}(\boldsymbol{\Delta}_j^{(t-1)} - \boldsymbol{\Delta}_i^{(t-1)})^\top \mathbb{E}[\Pi_{ijt}^\top]\boldsymbol{\Delta}_i^{(t-1)}$$

$$+ \sum_{j,k\in[n]} W_{ij}W_{ik}(\boldsymbol{\Delta}_j^{(t-1)} - \boldsymbol{\Delta}_i^{(t-1)})^\top \mathbb{E}[\Pi_{ijt}^\top\Pi_{ikt}](\boldsymbol{\Delta}_k^{(t-1)} - \boldsymbol{\Delta}_i^{(t-1)})$$

$$\preceq \boldsymbol{\Delta}_i^{(t-1)^\top}\boldsymbol{\Delta}_i^{(t-1)} + \sum_{j\in[n]} \delta W_{ij}\boldsymbol{\Delta}_i^{(t-1)^\top}(\boldsymbol{\Delta}_j^{(t-1)} - \boldsymbol{\Delta}_i^{(t-1)})$$

$$+ \sum_{j\in[n]} \delta W_{ij}(\boldsymbol{\Delta}_j^{(t-1)} - \boldsymbol{\Delta}_i^{(t-1)})^\top\boldsymbol{\Delta}_i^{(t-1)}$$

$$+ \sum_{j,k\in[n]} \delta W_{ij}W_{ik}(\boldsymbol{\Delta}_j^{(t-1)} - \boldsymbol{\Delta}_i^{(t-1)})^\top(\boldsymbol{\Delta}_k^{(t-1)} - \boldsymbol{\Delta}_i^{(t-1)})$$

$$= \boldsymbol{\Delta}_i^{(t-1)^\top}\boldsymbol{\Delta}_i^{(t-1)} - \delta\boldsymbol{\Delta}_i^{(t-1)}\boldsymbol{\Delta}_i^{(t-1)^\top} + \sum_{j,k\in[n]} \delta W_{ij}W_{jk}\boldsymbol{\Delta}_j^{(t-1)^\top}\boldsymbol{\Delta}_k^{(t-1)}.$$

The second matrix inequality used the fact that if $A \preceq B$ then $C^\top A C \preceq C^\top B C$ for any $C$. The equality in the third step pulled out the terms which only depend on $i$ from the expressions and used our assumption (A4) that $\sum_j W_{ij} = \sum_i W_{ij} = 1$. Taking trace on both sides and using $\text{Tr}(AB) = \text{Tr}(BA)$ we can simplify the expression as

$$\mathbb{E}[\text{Tr}(\boldsymbol{\Delta}_i^{(t)^\top}\boldsymbol{\Delta}_i^{(t)})] \leq (1-\delta)\text{Tr}(\boldsymbol{\Delta}_i^{(t-1)^\top}\boldsymbol{\Delta}_i^{(t-1)}) + \delta\,\text{Tr}((\sum_{j\in[n]} W_{ij}\boldsymbol{\Delta}_j)^\top(\sum_{j\in[n]} W_{ij}\boldsymbol{\Delta}_j))$$

The lemma now follows by the definition of Frobenius norm $\|Z\|_F^2 = \text{Tr}(Z^\top Z)$. $\quad\square$

**Lemma 3** (Effect of mixing). *Assuming that $\mathbf{W}$ has a spectral gap $\rho$ as in* (A4) *and $\boldsymbol{\Delta}_i^{(t)} := \mathbf{X}_i^{(t)} - \bar{X}^{(0)}$, we have*

$$\frac{1}{n}\sum_{i\in[n]}\left\|\sum_{j\in[n]} W_{ij}\boldsymbol{\Delta}_j^{(t-1)}\right\|_F^2 \leq (1-\rho)\frac{1}{n}\sum_{i\in[n]}\|\boldsymbol{\Delta}_i^{(t-1)}\|_F^2\,.$$

*Proof.* Follows from standard mixing arguments such as in (Xiao & Boyd, 2004b). $\quad\square$

Averaging lemma 2 over the nodes $i$ and then applying Lemma 3 gives

$$\frac{1}{n}\sum_{i\in[n]}\|\boldsymbol{\Delta}_i^{(t)}\|_F^2 \leq (1-\delta)\frac{1}{n}\sum_{i\in[n]}\|\boldsymbol{\Delta}_i^{(t-1)}\|_F^2 + \delta\frac{1}{n}\sum_{i\in[n]}\left\|\sum_{j\in[n]} W_{ij}\boldsymbol{\Delta}_j^{(t-1)}\right\|_F^2$$

$$\leq (1-\delta+\delta(1-\rho))\frac{1}{n}\sum_{i\in[n]}\|\boldsymbol{\Delta}_i^{(t-1)}\|_F^2$$

$$= (1-\rho\delta)\frac{1}{n}\sum_{i\in[n]}\|\boldsymbol{\Delta}_i^{(t-1)}\|_F^2\,.$$

This proves the statement of Theorem I. $\quad\square$

# B Compressed optimization (Proof of Theorem II)

We will use two main results proved in the previous section about our consensus step: that the average is preserved (Lemma 1), and that every step is a contraction in expectation (Theorem I). Any consensus operator which satisfies these two properties directly ensures convergence of the stochastic optimization method by the proof technique of (Koloskova et al., 2020). In particular, this shows that we satisfy Assumption 4 of (Koloskova et al., 2020) with $p = \rho\delta$. Replacing $p$ with $\rho\delta$ in their Theorem 2 yields the desired rates. $\quad\square$

 ## C  Experimental settings

 Tables 3, 4 and 5 describe the implementation details of our experiments.

Table 3: Default experimental settings for Cifar-10/ResNet-20 (based on Koloskova et al., 2019a)

| | |
|---|---|
| Dataset | Cifar-10 |
| Data augmentation | random horizontal flip and random $32 \times 32$ cropping |
| Architecture | ResNet-20 |
| Training objective | cross entropy |
| Evaluation objective | top-1 accuracy |
| Number of workers | 8 |
| Topology | ring |
| Network $W_{ij}$ | 0.436 for neighbors $i, j$,  0.128 if $i = j$,  0 otherwise |
| | (optimized for largest spectral gap) |
| Data | reshuffled between workers every epoch |
| Batch size | $128 \times$ number of workers |
| Momentum | 0.9 |
| Learning rate | Tuned. PowerGossip uses the same as uncompressed centralized all-reduce. |
| LR decay | $/10$ at epoch 150 and 250 |
| LR warmup | Step-wise linearly within 5 epochs, starting from 0.1 |
| # Epochs | 300 |
| Weight decay | $10^{-4}$,  0 for BatchNorm parameters |
| Repetitions | 6, with varying seeds |
| Reported metric | Worst result of any worker of the worker's mean test accuracy over the last 5 epochs |

Table 4: Default experimental settings for WikiText-2 (based on Vogels et al., 2019)

| | |
|---|---|
| Dataset | Word-level WikiText-2 |
| Tokenizer | Spacy |
| Architecture | 3-layer LSTM |
| Training objective | cross entropy |
| Evaluation objective | cross entropy / perplexity |
| Number of workers | 16 |
| Topology | ring |
| Network $W_{ij}$ | $\frac{1}{3}$ for neighbors $i, j$,  $\frac{1}{3}$ if $i = j$,  0 otherwise |
| | (common settings, worked better for DPSGD than weights used for Cifar-10) |
| Data | Source text strictly divided into 16 equal chunks, always remain on worker |
| Batch size | $64 \times$ number of workers |
| Momentum | 0.0 |
| Learning rate | Tuned. PowerGossip uses the same as uncompressed centralized all-reduce. |
| LR decay | $/10$ at epoch 60 and 80 |
| LR warmup | Step-wise linearly within 5 epochs, starting from 1.25 |
| # Epochs | 90 |
| Weight decay | 0.0 |
| Repetitions | 2 |
| Reported metric | Worst result of any worker of the worker's mean test cross entropy over the last 5 epochs |

Table 5: Experimental settings for Consensus

| | |
|---|---|
| Number of workers | 8 |
| Topology | ring |
| Network $W_{ij}$ | 0.436 for neighbors $i, j$,  0.128 if $i = j$,  0 otherwise |
| | (optimized for largest spectral gap) |
| Data | $100 \times 100$ random normal data |
| | or 8 randomly selected $64 \times 64$ faces from (AT&T Laboratories Cambridge) |
| Objective | minimize $\frac{1}{8} \sum_{i=1}^{8} \left( \mathbf{X}_i^{(t)} - \bar{X}^{(0)} \right)^2$ |

 ## D  Convergence curves

 Below, we plot the convergence curves in terms of test accuracy, as a function of either gradient
 updates (epochs) or bits sent per worker. In all our experiments, we have used a fixed number of
 epochs and a learning rate schedule that is common for full precision centralized training. It is
 possible that experiments with high communication compression would benefit from more epochs or
 a slightly different learning rate schedule.

## D.1 ResNet-20 on Cifar-10

## D.2 LSTM on WikiText-2

# E The power spectrum of parameter differences

## E.1 LSTM Training

The plots below show the power spectra of parameter differences observed while training the LSTM (Appendix I). We train with 16 workers connected in a ring, using PowerGossip with 32 power iterations per gradient update. During training, we record the power spectra of the differences between the parameters of connected workers 0-1, 4-5 and 8-9 at 4 different training stages. Lines are averages of the spectra observed between the three worker pairs.

The power spectra change significantly over time, but at most stages, they show that a few singular vectors carry more weight than others. This structure can be exploited by PowerGossip with power iterations. Especially in early training, the power spectra are peaky. This phase has been observed to be critical for successful training of non-convex models (Frankle et al., 2020).

## E.2 Consensus

The effect of a peaky spectrum on PowerGossip shows in our *consensus* experiments. When we plot the spectra of parameter differences between neighboring workers at initialization, we see that faces from the Faces Database (AT&T Laboratories Cambridge) can be approximated better with a low-rank approximation than random normal matrices. This is the reason why, in Figure 1, PowerGossip with power iterations is more efficient per-bit than uncompressed gossip for this dataset.

## F   Changing rank vs changing # power iterations

PowerSGD (Vogels et al., 2019), the algorithm on which PowerGossip is inspired, control their compression rate by varying the rank of the low-rank approximations. While this strategy is effective in terms of quality, it requires their projection matrices to be orthogonalized at every step of power iteration, rather than normalized. This operation scales as the square of the approximation rank, and is reported to be the most expensive step of the algorithm. A second disadvantage of using a high rank is that the memory required to store previous low-rank approximations scales linearly with the rank as well.

In PowerGossip, we adopt an alternative approach where we use multiple rank-1 power iteration steps per gradient update instead of one step with higher accuracy. In the table below, we show that this alteration has no impact on the performance of our method, evaluated with a fixed budget of 90 epochs on WikiText-2 language modeling. For the same total communication budget, we reach similar test loss.

| Sent/epoch | PowerGossip rank | Num. power iterations | WikiText-2 test loss |
|---|---|---|---|
| 127 MB | 1 | 8 | 4.73 |
| 230 MB | 1 | 16 | 4.63 |
| 437 MB | 1 | 32 | 4.58 |
| | 2 | 16 | 4.58 |
| | 4 | 8 | 4.58 |
| | 8 | 4 | 4.58 |

# G   Hyperparameters

## G.1   Consensus

In Figure 1, we plot results obtained with two compressors in ChocoGossip Koloskova et al. (2019b), using 20 consensus step size parameters $\gamma$ ranging from $7.6 \times 10^{-5}$ to 1 on an exponential grid. The optimal hyperparameter depends on the compressor used.

## G.2   ResNet-20 on Cifar-10

The table below specifies the optimizer-specific hyperparameters that we used in our experiments. For our baselines DeepSqueeze and ChocoSGD, we use tuned hyperparameters from Koloskova et al. (2019a).

| Method | Learning rate $\eta$ | | Consensus rate $\gamma$ | | Modulo parameter $\theta$ | |
|---|---|---|---|---|---|---|
| | Tested | Used | Tested | Used | Tested | Used |
| All-reduce (baseline) | $\{0.8, 1.13, 1.6\}$ | 1.13 | | | | |
| Uncompressed DP-SGD | $\{0.8, 1.13, 1.6\}$ | 1.13 | | | | |
| Choco (top-1%) | $\{0.96, 1.2, 1.6\}^\star$ | 1.13 | $\{0.025, 0.0375, 0.075, 0.15\}^\star$ | 0.0375 | | |
| Choco (Sign+Norm) | $\{1.2, 1.6, 2.4\}^\star$ | 1.6 | $\{0.15, 0.2, 0.45, 1\}^\star$ | 0.45 | | |
| Moniqua (2-bit) | $\{0.1, 0.2, 0.4, 0.8\}$ | 0.4 | $\{0.01, 0.005, 0.0025, 0.0012\}^\dagger$ | 0.005 | $\{0.125, 0.25, 0.5\}$ | 0.25 |
| DeepSqueeze (Sign+Norm) | $\{0.24, 0.48, 0.96\}$ | 0.48 | $\{0.005, 0.01, 0.05\}^\star$ | 0.01 | | |
| PowerGossip (1 iteration) | | 11.3 | | | | |
| PowerGossip (2 iterations) | | 11.3 | | | | |

$\star$:   based on published tuned parameters and the tuning strategy from the authors of ChocoSGD (Koloskova et al., 2019a).

$\dagger$: the concensus step size was tuned after the other parameters, not in a full grid.

## G.3   LSTM on WikiText-2

The table below specifies the optimizer-specific hyperparameters that we used in our experiments.

| Method | Learning rate $\eta$ | | Consensus rate $\gamma$ | | Modulo parameter $\theta$ | |
|---|---|---|---|---|---|---|
| | Tested | Used | Tested | Used | Tested | Used |
| All-reduce (baseline) | $\{15, 20, 27.5, 35, 47.5\}$ | 47.5 | | | | |
| Uncompressed DP-SGD | $\{15, 20, 27.5, 35, 47.5\}$ | 47.5 | | | | |
| Choco (top-1%)$^\dagger$ | $\{47.5\}$ | | $\{0.01, 0.1, 0.2, 0.4, 0.8\}$ | | | |
| Choco (Sign+Norm) | $\{35, 47.5\}$ | 47.5 | $\{0.4, 0.6, 0.8, 1.0\}$ | 0.8 | | |
| PowerGossip ($\star$ iterations) | | 47.5 | | | | |

$\dagger$: did not converge. We did not report this result, as more tuning may help.

# H   Compared-to algorithm implementations

In the sections below, we describe the implementation details of the algorithms we compare to. We provide the code for our implementations on Github (after deanonimization).

## H.1   ChocoSGD

We implement Algorithm 1 of (Koloskova et al., 2019a), which differs slightly from Algorithm 2 in (Koloskova et al., 2019b), in that it executes consensus steps and gradient updates in parallel like DP-SGD.

We use three compressors in our experiments. As customary, we compress each tensor parameter of our neural networks separately.

- **Sign+Norm**   $Q(\mathbf{x}) = \text{sign}(\mathbf{x}) \cdot \frac{\|\mathbf{x}\|_1}{\text{length}(\mathbf{x})}$.   We confirm the author's observations that this compressor gives the best and most reliable results.

- **top-1%**  Let $p_{99}(\mathbf{x})$ represent the 99th percentile of coordinates in $\mathbf{x}$ by absolute value. Here

$$Q(\mathbf{x})_i = \mathbf{x}_i \text{ if } \mathbf{x}_i \geq p_{99}(\mathbf{x}), 0 \text{ otherwise.}$$

  To communicate the top 1% of a vector, we communicate 32-bit float values and 64-bit integer indices, following the authors.

- **SVD**  This low-rank compressor has not been used with ChocoSGD, but we have evaluated it because our proposed method is also based on low-rank compression. This compressor represents a matrix $X$ by $(X\mathbf{v})\mathbf{v}^\top$, where $\mathbf{v}$ is the (normalized) top right singular vector found by a Singular Value Decomposition (SVD).

## H.2  DeepSqueeze

We implement DeepSqueeze according to Algorithm 1 in (Tang et al., 2019), and use the same compressors described for ChocoSGD above.

## H.3  Moniqua

Because the 1-bit version of Moniqua (Lu & Sa, 2020) is derived from the 2-bit version with added BZIP compression, we focus on the 2-bit version. We implement the algorithm according to Algorithm 1 in (Lu & Sa, 2020). We use the same step size schedule $\{\alpha_k\}$ as for the optimizers we evaluated, and tune the a priori bound $\theta$ as a gobal constant, as suggested by the authors. As a stochastic rounding operator $\mathcal{Q}$, we quantize stochastically in an unbiased fashion to the points $\{-\frac{1}{2}, -\frac{1}{6}, \frac{1}{6}, \frac{1}{2}\}$. This yields $\delta = \frac{1}{3}$. Note that the modulo operator 'mod $B_\theta$' in the algorithm yields values between $-\frac{1}{2}B_\theta$ and $\frac{1}{2}B_\theta$.

# I  Parameters in architectures

See Table 6 and Table 7 for an overview of parameters in the models used.

Table 6: Parameters in the ResNet20 architecture and their shapes. The table shows the per-tensor compression ratio achieved by rank-1 PowerGossip with $r$ iterations.

| Parameter | Parameter shape | Matrix shape | Uncompressed | Compression |
|---|---|---|---|---|
| layer3.1.conv1 | $64 \times 64 \times 3 \times 3$ | $64 \times 576$ | 144 KB | $115/r \times$ |
| layer3.2.conv1 | $64 \times 64 \times 3 \times 3$ | $64 \times 576$ | 144 KB | $115/r \times$ |
| layer3.0.conv2 | $64 \times 64 \times 3 \times 3$ | $64 \times 576$ | 144 KB | $115/r \times$ |
| layer3.1.conv2 | $64 \times 64 \times 3 \times 3$ | $64 \times 576$ | 144 KB | $115/r \times$ |
| layer3.2.conv2 | $64 \times 64 \times 3 \times 3$ | $64 \times 576$ | 144 KB | $115/r \times$ |
| layer3.0.conv1 | $64 \times 32 \times 3 \times 3$ | $64 \times 288$ | 72 KB | $105/r \times$ |
| layer2.2.conv2 | $32 \times 32 \times 3 \times 3$ | $32 \times 288$ | 36 KB | $58/r \times$ |
| layer2.1.conv1 | $32 \times 32 \times 3 \times 3$ | $32 \times 288$ | 36 KB | $58/r \times$ |
| layer2.0.conv2 | $32 \times 32 \times 3 \times 3$ | $32 \times 288$ | 36 KB | $58/r \times$ |
| layer2.1.conv2 | $32 \times 32 \times 3 \times 3$ | $32 \times 288$ | 36 KB | $58/r \times$ |
| layer2.2.conv1 | $32 \times 32 \times 3 \times 3$ | $32 \times 288$ | 36 KB | $58/r \times$ |
| layer2.0.conv1 | $32 \times 16 \times 3 \times 3$ | $32 \times 144$ | 18 KB | $52/r \times$ |
| layer1.1.conv1 | $16 \times 16 \times 3 \times 3$ | $16 \times 144$ | 9 KB | $29/r \times$ |
| layer1.1.conv2 | $16 \times 16 \times 3 \times 3$ | $16 \times 144$ | 9 KB | $29/r \times$ |
| layer1.0.conv2 | $16 \times 16 \times 3 \times 3$ | $16 \times 144$ | 9 KB | $29/r \times$ |
| layer1.2.conv1 | $16 \times 16 \times 3 \times 3$ | $16 \times 144$ | 9 KB | $29/r \times$ |
| layer1.0.conv1 | $16 \times 16 \times 3 \times 3$ | $16 \times 144$ | 9 KB | $29/r \times$ |
| layer1.2.conv2 | $16 \times 16 \times 3 \times 3$ | $16 \times 144$ | 9 KB | $29/r \times$ |
| layer3.0.downsample.0 | $64 \times 32 \times 1 \times 1$ | $64 \times 32$ | 8 KB | $43/r \times$ |
| fc | $10 \times 64$ | $10 \times 64$ | 2 KB | $17/r \times$ |
| layer2.0.downsample.0 | $32 \times 16 \times 1 \times 1$ | $32 \times 16$ | 2 KB | $21/r \times$ |
| conv1 | $16 \times 3 \times 3 \times 3$ | $16 \times 27$ | 2 KB | $20/r \times$ |
| Bias vectors (total) | | | 6 KB | None |

Table 7: Parameters in the LSTM architecture and their shapes. The table shows the per-tensor compression ratio achieved by rank-1 PowerGossip with $r$ iterations.

| Parameter | Parameter shape | Matrix shape | Uncompressed | Compression |
|---|---|---|---|---|
| encoder | $28869 \times 650$ | $28869 \times 650$ | 73300 KB | $1271/r \times$ |
| rnn-ih-l0 | $2600 \times 650$ | $2600 \times 650$ | 6602 KB | $1040/r \times$ |
| rnn-hh-l0 | $2600 \times 650$ | $2600 \times 650$ | 6602 KB | $1040/r \times$ |
| rnn-ih-l1 | $2600 \times 650$ | $2600 \times 650$ | 6602 KB | $1040/r \times$ |
| rnn-hh-l1 | $2600 \times 650$ | $2600 \times 650$ | 6602 KB | $1040/r \times$ |
| rnn-ih-l2 | $2600 \times 650$ | $2600 \times 650$ | 6602 KB | $1040/r \times$ |
| rnn-hh-l2 | $2600 \times 650$ | $2600 \times 650$ | 6602 KB | $1040/r \times$ |
| Bias vectors (total) | | | 174 KB | None |

## J Experiment runtime and compute infrastructure

We have executed our deep learning experiments on Nvidia Tesla K80 GPUs on `n1`-series virtual machines on Google Cloud. The algorithms were implemented in PyTorch, and run using a custom build that includes MPI for decentralized communication. We refer to the supplemental code for additional details on our runtime environment.

For our LSTM experiments with 16 workers, we use 4 GPUs with 4 processes per GPU. The experiments took approximately 4 hours in this setup.

For our Cifar-10 experiments with 8 workers, use 2 GPUs with 4 processes each. Those experiments took around 1.5 hours.