[Reviews · NeurIPS 2020]

Review 1

Summary and Contributions: The authors extended PowerSGD algorithm for Decentralized Deep Learning. They provided the convergence properties of the new approach and gave some numerical experiments to assess its efficiency.

Strengths: -Proposing a low rank compression technique for decentralized learning.

Weaknesses: -The authors explored only one class of compression techniques for decentralized learning. It will be interesting to explore other compression techniques as well in this setting

Correctness: I did not check the proofs

Clarity: Yes

Relation to Prior Work: Yes

Reproducibility: Yes

Additional Feedback: It will be interesting to explore how other compression techniques will perform in this setting as well. ---After the rebuttal----- I read the other reviews and the author response. I decided to keep my score unchanged to 7.


Review 2

Summary and Contributions: Post-rebuttal update: I am happy with the authors' response to my question on the bounded variance assumption. I maintain that the paper should be accepted. ================ The authors take inspiration from a power-method based compression method for efficient communication in distributed optimization. Instead, they apply this idea to the ‘decentralized’ setting, where communication is limited to neighboring nodes on some network topology. A long known property of the power method is its lightness in terms of hyper parameter tuning. Indeed, the authors provide a convincing method which compresses model differences between neighboring nodes using the power method and they prove that it achieves state of the art convergence results on non-convex, convex and strongly convex objectives. Their experiments demonstrate that they can achieve the same performance as other methods but without the need to tune hyperparameters.

Strengths: - The idea is not new, but has not been applied in the decentralized context. Very well motivated endeavor in my opinion. - The paper is very well written and well organized. - The results improve upon recent rates given for other compression schemes. - The authors show that the rates are asymptotically dominated by a stochastic term that does not depend on the compression factor, establishing that compression is not the bottleneck, and one can (asymptotically) enjoy lighter communication for 'free'. - The fact that the method is easy to tune is bound to have a positive impact on large decentralized systems.

Weaknesses: There is one weakness in the paper: the bounded variance assumption. The authors assume that the gradient variance is uniformly bounded by some sigma^2 for any value of X (A3). This type of assumption has been commonly made in the literature. However is it quite strong, especially for strongly convex problems. More importantly, state of the art results do not make this assumption (Koloskova et al, 2020 from your references). Can you please discuss about this assumption and its validity in practice? Is there any way to extend your results to a setting where the gradient variance might be allowed to grow with the norm of the gradient, for example?

Correctness: As far as I can tell, by checking the analysis and by cross-referencing the corner cases with existing results from literature the results are correct.

Clarity: The paper is written very clearly. It was an informative and pleasant read.

Relation to Prior Work: To the best of my knowledge, no recent, relevant result is missing and some important classics are also cited. The proposed work is well-put into context by the discussion. The difference and novelty are clear.

Reproducibility: Yes

Additional Feedback: I think that the paper should be accepted. I would like to see some discussion about the bounded-variance assumption (see above).


Review 3

Summary and Contributions: This paper proposes a low-rank compression algorithm suitable for differential compression of decentralized models. The algorithm uses power iterations to achieve maximum information conversion per bit. Experiments showed that the proposed algorithm does not require additional hyperparameters in the experiment, and compared with the previous gradient compression methods, the proposed algorithm converges faster and has higher accuracy. Overall I think this is a good paper.

Strengths: The key to the method proposed in this paper is the low-rank compression algorithm, which uses a time-varying vector v and the neighborhood difference matrix to perform multiplication operations and achieve data compression, and this is a linear process. The proposed algorithm has good practical application value in decentralized deep learning model training. And the theoretical analysis of this paper is comprehensive.

Weaknesses: I have read the cited article of PowerSGD and compared with the submitted paper. I am left with impression that the improvement of the PowerGossip is kind of incremental, and the experimental comparasion with the PowerSGD is missing from Table 1 and Table 2.

Correctness: Yes

Clarity: Yes

Relation to Prior Work: More discussion of the dfference between the proposed method and the PowerSGD is preferable.

Reproducibility: Yes

Additional Feedback: 1. More experiment results for the effect of the different ranks on the accuracy and comparession rate is preferable. 2. Is the PowerGossip a special form of the PowerSGD mathmatically? And under what condition does the PowerGossip outperforms PowerSGD, and what is the tradeoff and restriction. I have checked the rebuttal, but I still have the concerns about the issues raised by other reviews , so I keep my review score.


Review 4

Summary and Contributions: POST-REBUTTAL: I read the authors' response and I would like to maintain my score. In the next submission, I encourage the authors to (1) more clearly highlight their contributions as compared to the literature, (2) Discuss the trade-off of using larger-rank approximations. The paper proposes a compression algorithm for decentralized consensus optimization, based on low-rank approximation. The method is based on neighboring nodes i and j exchanging rank-1 approximation of power iterates to approximate the consensus update X_i-X_j. The authors prove a convergence result where the compression rate is in multiplicative form with the spectral gap, and experimentally validate their results on several tasks.

Strengths: * The paper proposes a practical algorithm that is simple and easy to implement. * Both theoretical and experimental results are reasonably strong.

Weaknesses: The two weaknesses that I can see are: * Insufficient comparison with the literature. In particular, novelty with respect to existing work is not sufficiently discussed. To my knowledge, the reference Cho et al. proposes a similar algorithm, though not in the consensus setting. More generally, it is not clear to me how novel the proposed algorithm is, and in light of the existing literature, the contributions can possibly be marginal. * It is not clear why the authors chose to limit themselves to rank-1 approximations. I understand that the convergence results are general enough to cover the rank-k scenario, but it would have been nice to see the empirical effect of the approximation rank too. In many settings, one may not need to compress at such an extreme rate, and the additional computational and communication cost of, say a rank-10 approximation, might well be worth the improved convergence rate.

Correctness: The results seem correct as far as I can tell.

Clarity: The paper is well-written and the results are easy to follow.

Relation to Prior Work: This is one of the weaknesses of the paper, see "Weaknesses" section.

Reproducibility: Yes

Additional Feedback:

[Author Response · NeurIPS 2020]

We thank the reviewers for their insightful comments and encouraging feedback. We hope that the comments raised are
addressed adequately below.

## Relation to PowerSGD and GradZip (R3, R4)

PowerSGD (Vogels et al., 2019) and GradZip (Cho et al., 2019) are two similar algorithms for *centralized* distributed
optimization that are also based on low-rank approximation. These methods approximate the average gradient update
across workers. This global averaging operation requires a fully-connected network and prevents straightforward
application of these methods in a decentralized setting.

The key difference in the proposed PowerGossip algorithm is that it instead approximates *model differences* between
connected workers. PowerGossip effectively instantiates multiple independent copies of PowerSGD; one for each pair
of connected workers. In the special case of a fully connected network, PowerGossip would use a different projection
vector for each pair of workers, rather than a global one as in PowerSGD.

## Relation to other algorithms for decentralized learning (R1)

We compare our work to other compression algorithms for decentralized learning (Koloskova et al. 2020, Tang et al.,
2019). While those algorithms also support low-rank compression, PowerGossip especially leverages the linearity and
contractivity of the operation by directly compressing model differences. This avoids the introduction of additional
hyperparameters that plagues prior work.

## Bounded variance assumption (R2)

The relaxation of the bounded variance assumption follows easily using standard techniques (using e.g. (Koloskova et
al. 2020) as pointed out by the reviewer). We chose to use a stronger assumption to ease presentation since we believed
that such a relaxation yields no new insights. We will be happy to extend our analysis to the relaxed assumption setting.

## Varying the compression rank (R4)

Similarly to PowerSGD, PowerGossip supports ranks larger than 1. A Rank-n compression step requires the same data
transfer as n rank-1 steps, and those alternatives work equally well (see Appendix F). We opt for multiple rank-1 steps
as it avoids an expensive orthogonalization operation (Vogels et al., 2019). There could be a benefit of larger ranks in
latency-bound settings. We can highlight this trade-off in the manuscript.

[Meta-Review · NeurIPS 2020]

All reviewers agreed that this is paper contains novel contributions and should be accepted. The authors are urged to read and address to the extent possible the constructive comments they received.